# Patient-Derived Tumor Organoids for Drug Repositioning in Cancer Care: A Promising Approach in the Era of Tailored Treatment

**DOI:** 10.3390/cancers12123636

**Published:** 2020-12-04

**Authors:** Silvia Vivarelli, Saverio Candido, Giuseppe Caruso, Luca Falzone, Massimo Libra

**Affiliations:** 1Department of Biomedical and Biotechnological Sciences, University of Catania, 95123 Catania, Italy; silvia.vivarelli7@gmail.com (S.V.); scandido@unict.it (S.C.); giu.caruso97@gmail.com (G.C.); m.libra@unict.it (M.L.); 2Research Center for Prevention, Diagnosis and Treatment of Cancer, University of Catania, 95123 Catania, Italy; 3Epidemiology Unit, IRCCS Istituto Nazionale Tumori “Fondazione G. Pascale”, 80131 Naples, Italy

**Keywords:** Patients Derived Tumor Organoids (PDTOs), translational oncology, anti-cancer care, drug repurposing, living biobank

## Abstract

**Simple Summary:**

Currently, organoid research is having a growing impact in oncology. Tumor organoids, directly derived from patients’ specimens, can easily be expanded and cryopreserved. For that reason, they are becoming an indispensable ally in clinics for quicker diagnosis and prognosis of malignancies. Patient-derived cancer organoids are used as a platform to predict the efficacy of standard-of-care, as well as novel drugs. Therefore, this approach might be further utilized for validating off-label molecules, in order to widen the cancer care offer.

**Abstract:**

Malignancies heterogeneity represents a critical issue in cancer care, as it often causes therapy resistance and tumor relapse. Organoids are three-dimensional (3D) miniaturized representations of selected tissues within a dish. Lately, organoid technology has been applied to oncology with growing success and Patients Derived Tumor Organoids (PDTOs) constitute a novel available tool which fastens cancer research. PDTOs are in vitro models of cancer, and importantly, they can be used as a platform to validate the efficacy of anti-cancer drugs. For that reason, they are currently utilized in clinics as emerging in vitro screening technology to tailor the therapy around the patient, with the final goal of beating cancer resistance and recurrence. In this sense, PDTOs biobanking is widely used and PDTO-libraries are helping the discovery of novel anticancer molecules. Moreover, they represent a good model to screen and validate compounds employed for other pathologies as off-label drugs potentially repurposed for the treatment of tumors. This will open up novel avenues of care thus ameliorating the life expectancy of cancer patients. This review discusses the present advancements in organoids research applied to oncology, with special attention to PDTOs and their translational potential, especially for anti-cancer drug testing, including off-label molecules.

## 1. Introduction

Despite the tremendous progress made in oncology prevention and care, cancer is still one amongst the main causes of death globally [1]. Intrinsic tumor heterogeneity represents one of the major issues for a successful clinical management. Cancers may derive from all types of tissues and organs [2]. A tumor originating from a certain tissue might differ between patients, as well as within a same subject (both spatially and temporally). Importantly, tumor heterogeneity often drives the rise of acquired resistance to therapy and recurrence [3].

It is necessary to develop novel preclinical or para-clinical approaches, in order to find new insights into cancer clonal evolution mechanisms, as well as providing personalized approaches able to defeat occurring therapy resistance and tumor relapse [4]. Patient-derived Tumor Organoids (PDTOs) represent a unique opportunity for basic and translational studies, thanks to their intrinsic applicability to patients [5].

The history of organoids is relatively recent, but with a surge in scientific interest worldwide, given their immense translational potential [6]. Why that? Because there is a meaningful promise inside a protocol by which a single human cell with stemness features, when posed in the right context, it is possible to generate an entire organ-in-a-dish. Animal cancer models are widely used in medical research, but they are costly and are limited by species-related differences, often demonstrating a lack of translational capability [6]. Moreover, there is a growing public reluctance to use animal drug-testing, and organoids might represent a human-relevant cutting-edge alternative for testing drug safety and efficacy [7,8]. On the other end, human two-dimensional (2D) cell culture is a valid platform for high-throughput drug screening, but it misses the overall complexity and heterogeneity of the original tumor [9].

In oncology, organoids can overcome the flaws of both animal models and 2D tumor cell lines. In fact, 3D human organoids are species-specific. Also, they bear a greater structural and functional complexity, compared to 2D cultures, although still representing a reductionist in vitro approach, and thus, allowing controlled experimental conditions. Additionally, organoids maintain a genetic stability when expanded, contrariwise to 2D cultures [10].

Organoids are made by a number of different cell lineages, including stem and differentiated cells; overall recapitulating the whole tissue architecture. They originated from the discovery of Programmed Stem Cells (PSCs), which possess the capability of reconstructing a tissue in vitro, overall recapitulating a specific germ layer [11,12,13]. Presently, organoids can be obtained, either from embryonic, induced or adult-resident stem cells [12,14,15,16,17]. When grown in specific conditions, cells with stemness, can divide and differentiate, thereby forming a nearer-physiological 3D miniaturized representation of a tissue within a dish [12,18,19]. For that reason, organoids represent a novel platform to study developmental organogenesis and tissue homeostasis [20]. Additionally, organoids represent an important tool for disease modeling. Key applications of organoids in medicine are: (1) Repair and regeneration of damaged tissues/organs; (2) elimination of altered/overgrowing tissues (such as in cancers) [21]. 

The first example of organoids used as disease modeling is represented by cystic fibrosis, which is a genetic disease caused by mutations to the cystic fibrosis transmembrane conductor regulator (CFTR) gene which encodes an ion channel protein [22]. This finding allowed the development of a time- and cost-effective rectal organoid-based drug screening procedure, which are applicable to patients [23,24]. For a recent review regarding the wide use of organoids as disease model platform, see [25].

In the following paragraphs the importance of organoid technology in both cancer biology and care will be presented, with particular attention to the PDTOs, which are directly derived from patients’ cancer specimens, thereby becoming a pivotal ally in translational oncology. Importantly, the current use of PDTOs for anti-cancer drug screening will be described, including their employment for drug repositioning in cancer care.

## 2. The Importance of Organoids in Oncology

The current impact of organoids in oncology is really profound. Organoid technology has been successfully applied to cancer research, where such 3D structures are often considered as proper mini-organs [26]. Indeed, organoids can be imagined as accelerators of medical research and personalized healthcare, as they represent a way to study human physiology and pathology with less ethical concerns, if compared to the animal models [27]. Three-dimensional organoids present advantages and disadvantages, both summarized in Figure 1. As great advantage, organoids are a 3D miniaturized representation of a multicellular tissue/organ, therefore they constitute a controllable reductionist physiological/pathological system [28]. Moreover, they are stable in culture for long time, they can be expanded and cryopreserved, for these reasons, they can be employed for compound screening purposes [29]. As a disadvantage, organoids lack the in vivo complexity (including the immune system component and the blood vessels network). Additionally, given their stem cell origin, organoids need specific growth media enriched in growth factors and extracellular matrix (ECM) components to expand [29]. Finally, if overgrown, the inner cells might lose their contact with nutrients and consequently die (apoptosis, necrosis) or senesce [30]. Kim and collaborators elegantly summarized both advantages and disadvantages of human organoids in their recent work [12].

Cancer organoids have been obtained by using different strategies. They may be generated from human healthy tissues (starting from adult or pluripotent stem cells), and are further engineered with mutational changes to oncogenes or tumor-suppressor genes—through the use of gene editing techniques, including the revolutionary clustered regularly interspaced short palindromic repeats (CRISPR)-Cas technology [31].

Alternatively, cancer organoids may be directly established from patients’ cancer biopsies, as well as additional patients’ specimens, including cancer cells isolated from needle biopsies, blood, urine or bronchial lavage [32]. The organoids that have originated from patients’ cancerous materials are named PDTOs and they can be obtained starting from several cancerous tissues, such as the liver, pancreas, stomach, intestine, prostate, breast, bladder, nervous system and others [33].

Normal tissue-derived organoids with genetic manipulation can help basic cancer research. They can be used to model the multistep transformation initiation process, the progression of carcinogenesis, cancer cells clonal evolution and selection [34,35]. Additionally, normal organoids are pivotal in comparative studies, as demonstrated by several recent findings regarding the role of Vitamin D and its active metabolite calcitriol in gut homeostasis and malignant transformation [36,37]. Finally, such 3D cultures can be used to perform toxicity studies, as demonstrated by Shinozawa and colleagues, who recently developed a human liver organoid based screening model for liver toxicology, which might help to study drug-induced liver injury and facilitate compounds optimization and screening [38].

On the other end, PDTOs can be generated from different subtypes and grades of cancers, and they represent a novel tool to study cancer metastasization processes [39,40]. Ultimately, PDTOs are becoming a pivotal in vitro complementary tool to support oncology research in clinics. In fact, PDTOs represent the novel frontier for defeating cancer heterogeneity, given their potential to be utilized as patient-specific substrate to personalize the anti-cancer therapy [41]. This latter application has a profound impact on screenings of novel or repurposed drugs, in order to design more tailored-to-patient therapies. A summary of the two types of organoids employed in oncology is reported in Figure 2.

The history of organoids started with several seminal observations made by Eiraku and colleagues and published in two works respectively, in 2008, and 2011. The authors characterized the mechanism by which embryonic-stem-cell-derived tissues spontaneously form 3D cup-structures in vitro [42,43]. Contemporarily, the landmark study from Sato et al., in 2009, described for the first time how a single stem cell - isolated from mouse small intestine - might give origin to a self-organized organoid with crypt-villi structure. The organoid took shape when grown in the presence of specific growth media, enriched in selected growth factors and hydrogel rich in ECM proteins [19]. In 2011, the developed culture protocols were further adapted by the same group to grow colon adenomas and adenocarcinomas [44,45]. These works represent an important milestone, which highlighted the concept that “there is no inherent restriction in the replicative potential of adult stem cells (or a Hayflick limit) ex vivo” [44,45]. Since then, there was a boom in organoid research, leading to the establishment of long-term cultures of normal, as well as tumor organoids from a plethora of different tissues.

In 2013, Lancaster et al. were able to generate a human pluripotent stem cell-derived cerebral organoid culture recapitulating the features of cortical development and useful tools to study brain development and cerebral disorders, as further corroborated by several subsequent studies [46,47,48,49,50]. In 2014, Karthaus et al. and Chua et al. separately obtained a fruitful long-term expansion protocol for prostate cell organoids which recapitulated prostate gland architecture [51,52]. 

In 2015, Boj and colleagues established culture conditions for organoids derived from normal and neoplastic murine and human pancreas tissues [53]. In the same year, Kessler and colleagues obtained long-term, Notch and Wnt dependent, organoids from human fallopian tubes [54]. Whereas, Bartfeld et al., cultured long-term human gastric organoids, starting from human gastric stem cells. Importantly, they applied the model to study *H. pylori* infection in vitro [55]. In 2016, Hubert et al., established tumor organoids from human glioblastomas, capable of recapitulating the tumor heterogeneity of parental tumors [56].

In 2017, Turco et al. established the growth conditions for hormone-responsive long-term organoid cultures of human healthy, as well as malignant, endometrium [57]. In the same year, Broutier and collaborators, extended this finding to human primary liver cancer-derived organoid cultures, including hepatocellular carcinoma, cholangiocarcinoma and the two types combined [58].

In 2019, Boretto and colleagues, cultured long-term organoids from a broad spectrum of endometrial pathologies, including endometrial cancer. These organoids faithfully reproduced the original lesion, when transplanted in vivo, thus, providing a powerful research model and drug screening tool [59]. The same year, Schutgens and collaborators developed human primary kidney tubular epithelial organoids, named tubuloids, given their resemblance to nephrons. Importantly, tubuloids were established also when stem cells were isolated from human urine, representing an applicable 3D model of infectious, malignant or hereditary kidney diseases [60]. In 2019, Sachs et al., obtained long-living organoids from human airway of healthy subjects, as well as patients affected by cystic fibrosis, infectious diseases or lung cancer [61].

The experimental readout techniques, currently used with organoid cultures, are growing at a fast pace. They vary from the classical fluorescence imaging, to flow cytometry, to survival analyses, to live cell imaging [62]. A recent advance consists in the use of optical metabolic imaging, which represents a real-time measurement tool helping especially organoid-based toxicology studies [63]. Additionally, molecular biology characterization techniques, including sequencing and all the essential multi-omics analyses, even at single-cell level, became very important as general outcome measurements in organoid science [64]. Due to experimental advances, at present, we are able to compare therapy outcomes observed in patients with drug efficacy results obtained in a dish with organoids. The predictive potential of patient-derived organoids in oncology led to their current use in clinics as complementary tool to the classic genetics, for directing the therapy choice.

## 3. History and Applications of PDTOs in Translational Oncology

PDTOs derived from tissue or tumor specific stem cells resemble the 3D cellular structure and maintain the genetics of the original tumor, in vivo, including its intrinsic heterogeneity. They also retain both structure and function of patient’s tissue of origin. Additionally, they have long-term expansion capability [9]. Therefore, PDTOs bridge the gap between 2D cultures and in vivo models in the pre-clinical research [41]. 

Overall, PDTOs bear the potential to translate from bench to bedside. In fact, PDTOs-based ex-vivo drug screenings are currently used in clinics to test drug efficacy and safety, and therefore, to shape the therapy around the patient. Additionally, PDTOs can be expanded into small volume dishes and they can be cryopreserved, thus allowing to build a living biobank [65].

Over the past several years, the scientific community has been attempting to unify protocols to obtain, expand, freeze and thaw normal and diseased organoids [66]. The medium composition is crucial, as well as the specific enrichment in terms of lineage-specific growth factors. Overall, the medium composition ensures both reliability and reproducibility of the results. Organoids can be grown with scaffold or scaffold-free. Scaffolds resemble the ECM, while scaffold-free might be droplets of culture medium or air-liquid interface [66]. 

Like all organoids, an intrinsic limitation of PDTOs is that they cannot recapitulate the whole organism interaction, such as: tumor-stroma, tumor-microenvironment (including microbiota), tumor-endothelium, tumor-multiorgan. In fact, PDTOs lack the architecture of an entire organ, including the surrounding normal tissue [5]. They lack a functional immune system and a whole-body axis, with all the intercellular networks. Moreover, PDTOs lack vasculature. Hence, they cannot grow too large, in order to allow an efficient diffusion of nutrients, as well as to avoid hypoxia and necrosis of the inner areas [67]. Finally, the organoid generation process bears an intrinsic cell-selection process, which might be taken always into account [68].

Current organoid research is trying to overcome these multiple limitations, with the help of old and novel technologies. For example, PDTOs can be: Genetically edited, co-cultured with immune cells, stromal cells, microinjected with microbes [68]. Moreover, PDTOs can be transplanted in humanized mice, obtaining patient-derived xenografts (PDX), thus, providing the body axis complexity [69]. Finally, a pivotal role is played by bioengineering, with the development of novel revolutionary 3D bioprinting and bio-fabrication materials. Due to these novel supports, organoids-on-a-chip might be the future to offer a PDTO that is better interconnected with other cell types, closely mimicking the body axis and rebuilding the entire-organism microfluidics [70,71,72,73,74]. The overall features and uses of PDTOs in oncology are reported in Figure 3.

Historically, Hans Clevers’ laboratory can be considered amongst the first to pioneer PDTOs research. As described above, the seminal paper about 3D organoid is from 2009, when Sato and colleagues discovered the in vitro conditions by which a single Lgr5+ stem cell from mouse small intestine could build a 3D self-organizing crypt-villus organoid. The organoids were able to grow and be passaged in the absence of a mesenchymal niche, within an ECM-based growth medium, in the presence of a defined cocktail of growth factors [19]. Two years later, in 2011, the same authors demonstrated that similar culture conditions were enough to establish such epithelial organoids from human small intestine, colon, esophagus, as well as from the neoplastic counterpart adenoma and adenocarcinoma, posing the bases to the development of PDTOs research [44].

From Clevers lab as well, the research of Boj and collaborators published in 2015 demonstrated that organoids may be established also from normal and neoplastic murine and human pancreas tissues [53]. In the same year, Huang and collaborators further demonstrated that it was possible to generate stable PDTOs from freshly resected ductal pancreatic tumors. PDTOs retained patient-specific physiological changes, such as hypoxia, oxygen consumption, epigenetics. Moreover, they showed to preserve differences in sensitivity to histone methyltransferase inhibitors. This important finding paved the way to the concept that PDTOs might be used for drug screening to identify patient-tailored therapy strategies [75].

In order to study the intrinsic tumor heterogeneity, in 2018, Roerink and co-authors sequenced single-cell level colorectal cancer patients’ clonal organoids, each derived from a single cancer stem cell. The authors confirmed the presence of both intra tumor diversification and differential drug response in closely related clones, thereby, suggesting that resistance may appear late in tumorigenesis [76].

In 2019, Ooft and collaborators demonstrated that treated PDTOs from metastatic lesions matched patients therapy response in clinics, thereby, showing that PDTOs can predict the efficacy of a therapeutic approach and assisting clinicians to select the best chemotherapy regimen to administer to a certain patient [39]. On the same line, in 2019 Pash and colleagues demonstrated that PDTOs treatments are predictive of the treatment-sensitivity found in patients subjected to chemotherapy or radiotherapy [77]. In 2020 Nanki et al., obtained long-living ovarian cancer PDTOs that recapitulated the mutational landscape of the primary tumors. The obtained organoids have been successfully used to perform personalized drug sensitivity and resistance testing with about thirty FDA approved drugs, demonstrating the feasibility of using PDTOs to predict drug responses before the administration to patients [78]. 

Organoids can be used as a tool to study the effect of gut microbiota on colorectal cancer development, as demonstrated by Pleguezuelos-Manzano et al. in 2020, which injected genotoxic pks+ *E. coli* in the lumen of CRC PDTOs, thus discovering that such bacteria might determine a mutational process in the exposed organoids, being pro-tumorigenic [79].

As said, PDTOs can be expanded in culture. Consequently, many PDTOs can be derived from a single patient. This feature makes PDTOs suitable for larger drug screening, very useful to build a personalized anti-cancer therapy. Importantly, as described above, PDTOs may be successfully frozen and thaw [80]. Newly developed cryopreservation protocols are posing the bases to build living libraries of organoids, available for the matching patient, virtually at any moment during his medical path [80]. Meaning that, if a first line of anti-cancer treatment results in recurrence or relapse, cryopreserved PDTOs may thaw ad hoc and be used for the testing of additional molecules. The PDTOs are predictive tools in first, second and third lines of treatment and they might accompany cancer patients during their whole care pathway [81].

In 2014, Gao and colleagues successfully generated long-term prostate cancers PDTOs, from both biopsies and tumor circulating cells, that recapitulate the molecular diversity of prostate cancer subtypes. This result paved the way to build a living library of PDTOs with the idea that organoids deriving from a single patient can be preserved [82]. Subsequently, in 2015, van de Wetering and colleagues established for the first time a PDTOs living biobank from 20 different colorectal cancer patients. The PDTOs recapitulated the CRC tumor heterogeneity typical of CRCs. Hence, expanded on-demand, PDTOs may be suggested to complement drug screening and to find gene-drug associations therefore helping to tailor the anti-cancer treatment to the specific case [83]. The same year, Weeber and colleagues efficiently established organoids cultured from biopsies of human CRC metastases. The authors verified that PDTOs genetically represented the metastasis they were derived from. These organoids might constitute a drug-screening platform to help to personalize the anti-cancer therapy, based on the genetic profiling metastasis-derived organoids [84].

In 2016, Fujii and colleagues generated a living biobank of CRC organoids, from patients with heterogeneous forms of the neoplasm, from both primary and metastatic sites. The biobank serves to perform accurate genotype-phenotype analyses at a single-patient level, thereby helping in the choice of a tailored therapeutic approach [85].

Another important work that helped characterize the features of PDTOs is from Seino and colleagues, who in 2018, established a pancreatic tumor organoid library from 39 patients. They altered the growth factor availability and/or genetically modified the organoids, overall changing the response to niche specific factors such as Wnt or R-spondin. Using this experimental approach, the authors found that, while the tumor progresses, those factors become non-essential for PDTOs survival [86]. The same year, Sachs and colleagues developed a living biobank also from breast cancer PDTOs, which was demonstrated to be a valid drug screening tool, faithfully recapitulating the tumor heterogeneity in patients [87].

Additionally, in 2018, Vlachogiannis and colleagues established a living biobank of PDTOs from metastatic pretreated colorectal and gastroesophageal cancers. The matching of clinical outcomes and molecular profiling of PDTOs demonstrated that such 3D models may help to tailor a personalized treatment for patients who showed relapse and recurrence [88]. 

Moreover, the same year Lee and colleagues developed a living biobank of bladder cancer PTDOs, showing that such organoids recapitulate histopathological and molecular diversity of the original tumor. Analyses of drug response using bladder tumor organoids showed changes associated with treatment resistance and specific patients’ responses, thereby, representing a faithful model system for studying tumor evolution and treatment response in the context of precision cancer medicine [89].

In 2018, Tiriac and colleagues generated a pancreas PDTOs library. The derived organoid-based gene expression signatures of chemo-sensitivity efficiently predicted patients’ response to chemotherapy, thus enabling a prospective therapy selection [90]. The same year, Yan and colleagues developed a living biobank of human primary gastric cancer organoids and metastases from patients. The PDTOs were used to screen novel treatment at larger scale, revealing sensitivity to unexpected drugs, including Napabucasin, Abemaciclib, and the ATR inhibitor VE-822. Therefore, this biobank may provide a useful tool for precision cancer therapy [91]. The same year, Beshiri and colleagues developed a PDTOs biobank of advanced prostate cancers, and observed that the response to PARP inhibitor registered in vitro, as well as in PDX, was similar to the response observed in patients, thus evidencing the screening potential of PDTOs [92].

In 2019, Driehuis and collaborators developed long-term culture conditions for human mucosal PDTOs. The authors generated a living biobank from 31 head and neck squamous cell carcinoma (HNSCC), which overall recapitulated morphology and genetics of HNSCC in patients [93]. In 2019, Mullenders and colleagues optimized the culture conditions to obtain organoids from mouse and human urothelial cancers, from both resected tumors and biopsies. Such organoids could be passaged for long time. In particular, the PDTOs organoids from 53 patients were used to constitute a living biobank [94]. The same year, Kopper and colleagues efficiently differentiated and propagated ovarian cancer organoids from patients’ biopsies. They established 56 PDTOs starting from 32 patients, overall recapitulating ovarian cancer heterogeneity observed in clinics. PDTOs were efficiently used for drug-screening assays and were stored as a living biobank entirely available for the research community [95].

In 2020, Yao and colleagues established a living biobank of rectal cancer PDTOs, treated with neoadjuvant chemoradiation. Importantly, the co-clinical data demonstrated that rectal cancer organoids consistently recapitulated the pathophysiology and genetic changes of corresponding tumors in patients. Therefore, these PDTOs may represent an important tool for clinicians to help the diagnosis and prognosis of rectal cancer [96].

The same year, glioblastoma patients derived organoid cultures have been successfully developed by Jacob and colleagues. Such PDTOs recapitulated the histological features, cellular diversity, gene expression, and mutational profiles of their corresponding parental tumors. They also established a living biobank, useful as resource for basic and translational research, including drug screenings [97].

Moreover, Calandrini and colleagues efficiently originated tumor and matching normal kidney organoids from over 50 children with different subtypes of kidney cancer and constituted a publicly available living biobank. The PDTOs retained the properties of native tumors, therefore demonstrating to be an in vitro source to perform basic cancer research, as well as drug screenings for personalized care [98].

All the studies hereby reported highlight the increasing value of PDTO-based research, which represents a beneficial in vitro ally to study tumor biology, as well as to support translational research. The seminal studies on tumor organoids (see Section 2), as well as the up-to-date studies on PDTOs and their applications in basic science and translational oncology (described in this Section 3 and below in Section 5), are summarized in Table 1 (enclosing all the relevant aspects of each study mentioned).

## 4. Current Ongoing Clinical Studies Using PTDOs in Cancer Care

Currently, about 60 clinical trials employ PDTOs as a powerful complementary tool in cancer care. Table 2 summarizes open clinical studies deposited in the main registries [101,102].

PDTOs are established from many solid tumors of different origin, including the bile duct, bones, breast, esophagus, stomach, intestine, pancreas, lung, nasopharynx, prostate, ovary, reproductive organs. Moreover, one study attempted to derive PDTOs directly from bone-marrow myeloma cells through the use of a scaffolding synthetic biomaterial as a growing support (NCT03890614). Importantly, the current clinical studies involve the production of PDTOs, either from primary tumors or from metastases. 

In relation to the indicated primary outcomes of the studies, 16 trials aim to establish long-living PDTOs to perform association studies, in order to overall validate the genotypic and phenotypic correspondence between PDTO and tumor in patients. The final goal will be to find a signature of the specific malignancy and eventually identify novel biomarkers. In 39 clinical studies, once the organoid cultures are efficiently established and analyzed, PDTOs will be further used for drug screening of standard-of-care, as well as novel anti-cancer molecules. The goal will be to match individual patient outcome with PDTO drug-response, in order to assess whether the PDTOs can be efficiently used as in vitro predictive tool. Interestingly, only 4 trials currently include the generation of a living-PDTOs biobank within the main trial outcomes.

## 5. PDTOs and Drug Repositioning: A New Hope in Cancer Care

As described above, neoplasms bear an intrinsic heterogeneity. In light of that issue, the combination of multiple anti-cancer treatments might offer the opportunity to attack the tumor on several fronts, thus increasing the overall treatment success rate [103]. Despite this possibility, one of the main problems with malignancies are recurrences. The failure of the first-, second-, third-line treatments often leaves the oncologist without any available therapeutic option [103].

The compassionate use of drugs implies repurposing a known molecule, used for a primary non-oncology application, in order to offer an alternative, more effective, therapeutic option to cancer patients [104]. Great advantages of drug repurposing are their time- and cost-effectiveness. This is due to the fact that normally a re-purposed drug has already passed a large number of toxicity tests, with reduced risks in terms of safety issues and adverse events. For drugs used off-label, the time of development is halved, from 10 to 5 years. Consequently, the whole production costs are significantly reduced [105].

Many off-label molecules are currently studied for cancer treatment, including: metformin, aspirin, statins, estrogen receptor modulators, beta-blockers, antipsychotics, antidepressants and antimicrobials [106,107,108,109,110,111,112]. Several success stories have been reported for repositioned drugs, as for thalidomide used for multiple myeloma treatment [113,114]. 

The approaches to discover and assess the hidden anti-cancer potential of old drugs range from the artificial intelligence-based screenings to randomized clinical studies [115]. Figure 4 describes in details the steps involved in the drug repurposing process. The first step consists in the identification of the compound which normally lasts up to two years and consists in a mixture of different in silico approaches (including network analyses, molecular docking and molecular similarity studies and data mining) [116]. In particular, data mining is of key importance, as implies crossing searches and observations from multiple databases and drug repositories, literature search and connections between clinical retrospective studies results [105]. 

Once novel drug-disease connections are positively identified, the second important step consists in the experimental validation, which is performed with both in vitro and in vivo approaches [117,118]. In silico identification plus in vitro and in vivo validation together represent the discovery phase of the drug repurposing. 

If the discovery phase has been successful, the repurposed compound can be tested in patients. This third step, given the already existing history of toxicology and safety studies, normally starts with Phase II or III trials, which shorten the overall clinical validation time-frame to up to 5 years [119]. Once the repurposed compound passes the clinical validation, it can be registered and produced to the market (Figure 4) [120].

As for non-cancer drugs, therapies administered to cure a certain malignancy can be repurposed off-label for certain other tumors. The off-label prescription is commonly practiced in cancer therapy and often offered to patients who have no alternative options, to maximize the likelihood of a favorable benefit–risk ratio [121]. In this case there is no need for a discovery and validating process, but a prescription as adjuvant or palliative care, on a case-by-case basis [122].

In the Era of precision medicine, PDTOs represent a preclinical model that faithfully recapitulates the heterogeneity of the tumors, thus representing a realistic and effective drug screening model [25]. Given their effectiveness in reproducing tumor heterogeneity, and supported by the growing use in drug screenings, PTDOs represent an excellent substrate to perform repurposed drug screenings. This in vitro tool might represent a faster way to validate whether there is the necessity to undertake or not clinical trials, cutting the costs of off-label drug development and validation [25]. As a recent example in which drug repurposing is becoming pivotal is represented by the current ongoing COVID-19 pandemic, where organoids demonstrated to be an excellent in vitro screening platform [123]. Interestingly, organoid-based drug-repurposing screenings “went viral” already few years ago, during Zika virus outbreak in 2015 [124].

As reported above, PDTOs response showed large correspondence with cancer patients’ outcome [125,126,127,128,129]. Additionally, PDTOs-based drug tests have been further improved to allow medium and high throughput screening for old and new anti-cancer molecules. The potential importance of PDTOs in drug repositioning is summarized above in Figure 4. PDTOs are pivotal tools in two main phases of the drug repurposing process. Firstly, in the compound validation phase, to test in vitro both the toxicity and the efficacy of a given compound (new or repurposed). Also, PDTOs can be injected in immunodeficient mice, whereby PDX models can characterize the compound in vivo [33,130]. Secondly, as widely described in the sections above, PDTOs are currently employed in clinics as a screening platform to predict the efficacy of a given treatment, thus, tailoring the therapy around the patient. Therefore, PDTOs in this phase can be used as efficient complementary tool to compare with the clinical outcomes during drug repurposing studies (Figure 4) [25].

On this ground, in 2017, Pauli et al. developed a unique platform for precision medicine drug-screening in cancer patients combining PDTOs and PDXs, with the final aim of developing therapeutic options available for patients who fail to respond to the first-line, second-line and third-line treatments. The platform demonstrated to help to identify which somatic alterations to target, thus directing the choice for next-line therapy in cancer patients who develop resistance and relapse [131]. PTDOs sequencing or viability readouts are not the only available outcomes for drug-based screenings. The same year, Jabs and colleagues developed a microscopy-based assay to easily detect the viability response of patient-specific organoids to cytotoxic and cytostatic drugs, thus fastening the screening procedure [132].

In 2018, Lampis et al., optimized a 96-well plate platform for PDTO-based high throughput screening of 484 small-molecules targeting HSP90. Such PDTOs were established from cholangiocarcinomas patients’ biopsies. The long-term survival for this tumor is still poor and the aim was to search for novel therapeutics to ameliorate the positive outcomes [99]. 

More recently, in 2020, Narasimhan and colleagues developed a medium-throughput drug panel testing platform, based on organoids derived from peritoneal metastases of colorectal cancer patients, or peritonoids. This study demonstrated the feasibility of a PDTO-based platform for patients in urgent need for alternative treatment options [100]. Importantly, upon failure of standard care treatment, two out of 19 patients for whom PDTOs were generated, received treatments based on in vitro screening results. One of them had a partial response to the recommended treatment, despite being previously progressing upon standard care therapies [100]. 

Very recently in October 2020, Driehuis et al. described a robust unifying protocol to isolate and propagate in an efficient and reproducible way PDTOs for drug screening applications. The method unifies the most utilized and performing protocols currently available. A current issue is, in fact, to establish high-throughput organoid cultures, and this protocol allows researchers to efficiently scale down PDTOs for compounds screenings, up to a 384-well format [33].

The key examples reported above, highlight the importance of PDTOs in drug development and the potential applicability of such protocols to the drug repurposing process. Importantly, PDTOs represent the future of cancer research, as demonstrated by the fast-pace advancing results. As described in Section 4, about 60 clinical studies currently include PDTOs as complementary in vitro approach to evaluate tumor features and therapy response, as well as to tailor the treatments around the patient. On these bases, the high-throughput translational potential of PDTOs might be successfully applied to repurpose both non-cancer and cancer off-label drugs in oncology. 

Accordingly, the UK National Cancer Institute (NCI) is currently managing a key initiative called “the Human Cancer Models Initiative”, which aims to establish up to 1000 PDTOs cultures, to add them to the NCI’s library of tumor models [133]. Furthermore, many companies are now offering high throughput drug screening services with PDTOs and a range of outcome readouts, from morphology, to cell health, to sequencing and omics analyses [134,135,136]. In the nearer future, the PDTOs libraries and PDTO-based services will represent a valuable resource for translational cancer research, thus, contributing to the identification of novel cancer diagnostic markers, as well as the development of individualized patients’ therapeutic strategies and drug repositioning choices.

## 6. Conclusions

Since their discovery ten years ago, organoids became pivotal in oncology. In particular, the possibility of easily deriving PDTOs has opened a whole new avenue in cancer research. PDTOs are a tool to study tumor dynamics and they also represent a way to mirror the efficacy of anti-cancer therapy within a dish. For that specific feature, PDTOs represent a valuable tool for clinicians.

The research in PDTO field is growing at a fast pace and novel biomaterials enable 3D cultures to be obtained from non-solid tumors. Moreover, national healthcare systems, such as National Institute of Health (NIH). are currently investing in cancer organoid research, and PDTOs libraries will allow real-time high-throughput screen of old and novel drugs to tailor the therapy around each single cancer patient, thereby bypassing the issue of tumor-intrinsic heterogeneity. 

In light of all the presented perspectives, PDTOs may be applied also to drug repurposing in cancer care. They will help to validate off-label drugs, which in turn, will offer new opportunities as compassionate anti-cancer therapy options for patients in need of valid alternatives.

## Figures and Tables

**Figure 1 cancers-12-03636-f001:**
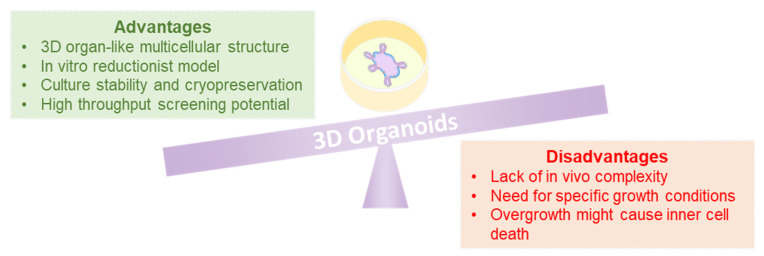
The balance of advantages and disadvantages of 3D organoids. As any experimental models, 3D organoids show both advantages (green box), as well as disadvantages (red box). Both sides might be taken into account when designing a study.

**Figure 2 cancers-12-03636-f002:**
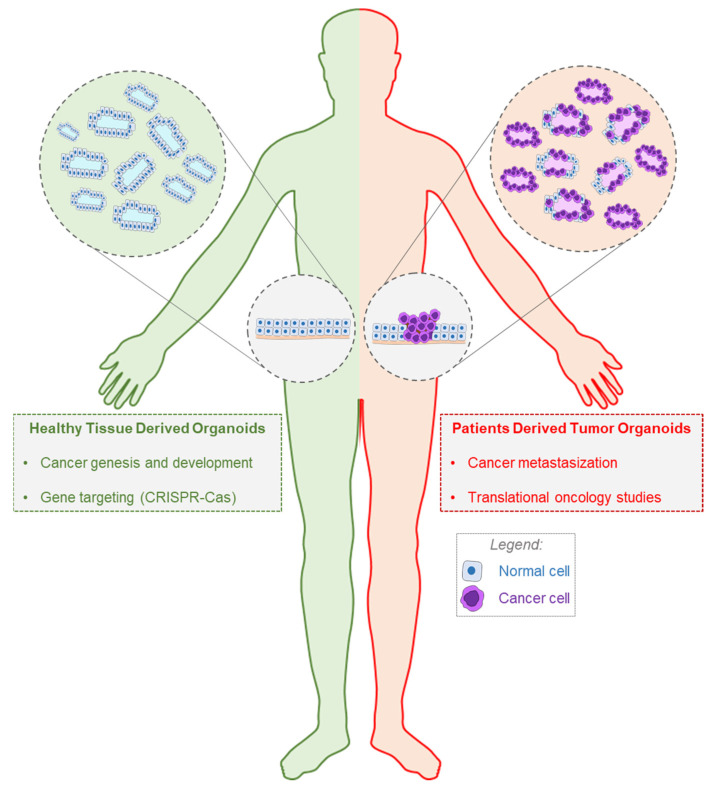
Types of organoids used in oncology. In cancer research, organoids either derived from healthy tissues (green box) or from cancer specimens (red box) can be employed. While, healthy organoids may be genetically modified and used to study cancer genesis and development (green box), Patients Derived Tumor Organoids (PDTOs) are valid tools to study metastasization and for translational oncology applications (red box).

**Figure 3 cancers-12-03636-f003:**
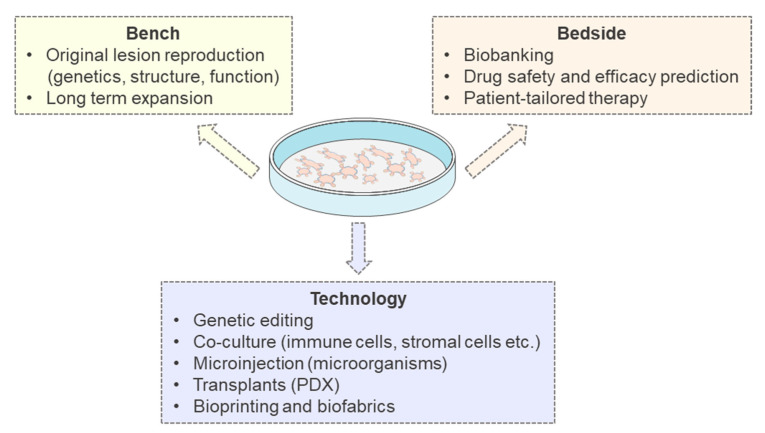
Translational importance of Patients Derived Tumor Organoids (PDTOs). PDTOs are useful at the bench (yellow box; they reproduce genetics, structure and function of the original tumor and they can be expanded long-term), at the bedside (orange box; they can be cryopreserved and thaw, they are a predictive tool in drug testing and can be used for personalized medicine). Moreover, organoid technology can be integrated (purple box; with gene editing, co-culture, microinjections, xeno-transplants, biomaterials). PDX, Patients Derived Xenografts.

**Figure 4 cancers-12-03636-f004:**
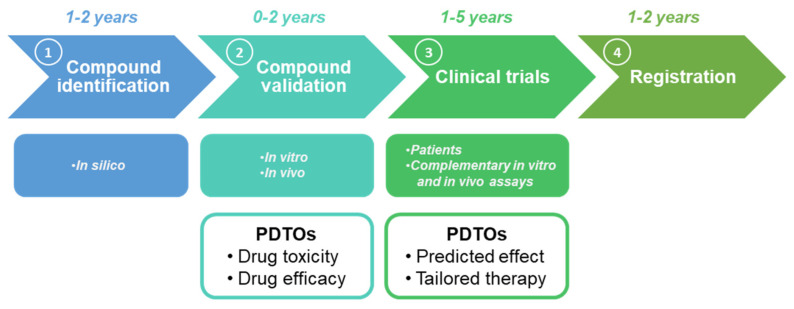
Patients Derived Tumor Organoids (PDTOs) and drug repurposing. The drug repositioning process is shorter than the drug discovery one and includes four phases: 1, Compound identification; 2, Compound validation; 3, Clinical trials; 4, Registration. PDTOs are useful tools during two phases: compound validation and clinical trials.

**Table 1 cancers-12-03636-t001:** Up-to-date cancer organoids and PDTOs preclinical and paraclinical studies.

Author	Year	Tissue(s)	Cell Type	Stem Cell	Drug Screening	Biobanking	PDX	Relevant Assays/Treatments	Study Potential(s)	Reference
Eiraku	2008	Cortical neurons	N	E	na	na	na	na	na	[42]
Sato	2009	Small intestine	N	A	na	na	na	na	na	[19]
Eiraku	2011	Retinal epithelium	N	E	na	na	na	na	na	[43]
Jung	2011	Colon	N, T	A	na	na	na	na	na	[45]
Sato	2011	Colon	N, T	A	na	na	na	na	na	[44]
Lancaster	2013	Brain	N	E	na	na	na	na	na	[46]
Chua	2014	Prostate	N, T	A	na	na	na	na	Biobanking, Drug screening	[52]
Gao	2014	Prostate, Blood	M	A	PI3-kinase pathway inhibitors, Everolimus, BKM-120	na	Y	na	Biobanking, Drug screening	[82]
Karthaus	2014	Prostate	N, T	A	na	na	Y	na	na	[51]
Bartfeld	2015	Stomach	N, T	A	na	na	na	*H. pylori*	Drug screening	[55]
Boj	2015	Pancreas	N, T	A	na	na	Y	na	na	[53]
Huang	2015	Pancreas	N, T	A	Histone methyltransferase EZH2 inhibitors	na	Y		Patient predictive screening	[75]
Kessler	2015	Fallopian tube	N	A	na	na	na	na	na	[54]
van de Wetering	2015	Colon	N, T	A	83 compound-library	Y	na	na	Drug screening	[83]
Weeber	2015	Colon	M	A	na	na	na	na	Drug screening	[84]
Fujii	2016	Colon	T	A	na	Y	Y	na	Drug screening	[85]
Hubert	2016	Brain	T	A	na	na	Y	na	Drug screening	[56]
Broutier	2017	Liver	T	A	Gemcitabine, Nutlin-3a, LGK974, ERK inhibitor (SCH772984)	na	Y	na	Drug screening	[58]
Turco	2017	Endometrium	N	A	na	na	na	na	Biobanking, Drug screening	[57]
Beshiri	2018	Prostate	T	A	Olaparib	Y	Y	na	na	[92]
Lampis	2018	Liver	T	A	484 compound-library (new small-molecules)	Y	Y	na	na	[99]
Lee	2018	Bladder	T	A	40 compound-library (ERK, MEK inhibitors)	Y	Y	na	Patient predictive screening	[89]
Roerink	2018	Colon	T	A	na	na	na	Single-cell sequencing	na	[76]
Sachs	2018	Breast	T, M	A	Afatinib, Gefitinib, Pictilisib, GDC-0068, AZD8055, Everolimus, Tamoxifen	Y	Y	Gene engineering CRISPR-cas9	na	[87]
Seino	2018	Pancreas	T	A	na	Y	Y	Gene engineering CRISPR-cas9	na	[86]
Tiriac	2018	Pancreas	T	A	Gemcitabine, Paclitaxel, Irinotecan, 5-Fluorouracil, Oxaliplatin	Y	na	na	Patient predictive screening	[90]
Vlachogiannis	2018	Stomach, Intestine	M	A	55 compound-library (FDA approved)	Y	Y	na	Patient predictive screening	[88]
Yan	2018	Stomach	N, T, M	A	37 compound-library (FDA approved and new)	Y	Y	na	na	[91]
Boretto	2019	Endometrium	N, T	A	Paclitaxel, 5-Fluorouracil, Carboplatin, Doxorubicin, Everolimus	Y	Y	na	na	[59]
Driehuis	2019	Oral mucosa	T	A	Nutlin-3, Niraparib, AZD4547, Everolimus, Vemurafenib, Alpelisib, Cisplatin, Carboplatin, Cetuximab	Y	Y	Herpes simplex virus, Human papillomavirus	Patient predictive screening	[93]
Kopper	2019	Ovary	N, T, M	A	Carboplatin, Paclitaxel, MK2206, AZD8055, Pictilisib, Alpelisib, Nutlin3a, Adavosertib, Gemcitabine, Niraparib	Y	Y	Gene engineering CRISPR-cas9	na	[95]
Mullenders	2019	Bladder	T	A	Epirubicin, Mitomycin C, Gemcitabine, Vincristine, Doxorubicin, Cisplatin	Y	na	Gene engineering CRISPR-cas9	Patient predictive screening	[94]
Ooft	2019	Colon	T, M	A	5-Fluorouracil, Capecitabine, Irinotecan, Oxaliplatin	na	na	na	Patient predictive screening	[39]
Pasch	2019	Various	T	A	5-Fluorouracil, Oxaliplatin, Ionizing radiation	na	na	na	Patient predictive screening	[77]
Sachs	2019	Lung, Broncho-alvolar lavage	N, T	A	Paclitaxel, Methotrexate, Crizotinib, Cisplatin, Nutlin-3a, Erlotinib, Alpelisib, Gefitinib	Y	Y	Respiratory syncytial virus	Drug screening	[61]
Schutgens	2019	Kidney, Urine	N, T	A	Cidofovir	na	na	BK virus, Organ-on-a-chip	Drug screening	[60]
Calandrini	2020	Kidney	N, T	A	Actinomycin D, Vincristine, Doxorubicin, Etoposide	Y	na	Single-cell sequencing	na	[98]
Jacob	2020	Brain	T	A	Ionizing radiation, Temozolomide, Gefitinib, Trametinib, Everolimus	Y	Y	CAR-T cells Co-culture	na	[97]
Nanki	2020	Ovary	T	A	23 compound-library (FDA approved)	na	na	na	na	[78]
Narasimhan	2020	Colon	M	A	87 compound pan-cancer library	na	Y	na	Patient predictive screening	[100]
Pleguezuelos-Manzano	2020	Colon	T	A	na	na	na	*E. coli*	na	[79]
Yao	2020	Rectum	T	A	Ionizing radiation, 5-Fluorouracil, and Irinotecan	Y	na	na	Patient predictive screening	[96]

PDX, Patient Derived Xenograft; N, Normal; T, Tumor (primary site); M, Metastasis; E, Embryo; A, Adult; Y, Yes; na, Not associated.

**Table 2 cancers-12-03636-t002:** Currently open clinical trials using PDTOs in cancer care.

Registry	Study ID	Origin	PDTOs Establishment	Drug Screening	Biobanking	Tumor Type	Enrollment	Location
Chinese CTR	ChiCTR1800016734	S, B	Y	N	N	Stomach	10	China
Chinese CTR	ChiCTR1800017767	S, B	Y	N	N	Ovary	120	China
Chinese CTR	ChiCTR2000034996	S, B	Y	N	N	Esophagus	50	China
US NIH	NCT02436564	S, B	Y	N	N	Liver, Pancreas	75	United Kingdom
US NIH	NCT02910895	S, B	Y	N	N	Soft Tissue	40	Netherlands
US NIH	NCT03140592	B	Y	N	N	Pancreas	300	USA
US NIH	NCT03952793	Ext B (Metastasis)	Y	N	N	Prostate	20	France
US NIH	NCT03990675	B	Y	N	N	Pancreas	50	Germany
US NIH	NCT04219137	S, B	Y	N	N	Stomach, Esophagus	120	Canada
US NIH	NCT04342286	S, B	Y	N	N	Kidney	20	China
US NIH	NCT04371198	S, B	Y	N	N	Rectum	20	USA
US NIH	NCT04478877	S	Y	N	N	Meningis	30	Hong Kong
Netherlands CTR	NL8956	S, B	Y	N	N	Larynx	20	Netherlands
Netherlands CTR	NTR6150	Blood Circ (Metastasis)	Y	N	N	Prostate	46	Netherlands
Netherlands CTR	NTR7286	B	Y	N	N	Breast	30	Netherlands
Thailand CTR	TCTR20200827007	B	Y	N	N	Pancreas	28	Thailand
Chinese CTR	ChiCTR1800018069	S, B	Y	Y	N	Breast	100	China
Chinese CTR	ChiCTR1900023682	S, B	Y	Y	N	Endometrium	20	China
Chinese CTR	ChiCTR1900027081	S, B	Y	Y	N	Nasopharynx	25	China
Chinese CTR	ChiCTR1900028000	B	Y	Y	N	Pancreas	50	China
Chinese CTR	ChiCTR2000028856	S, B	Y	Y	N	Stomach	59	China
Chinese CTR	ChiCTR2000028889	S, B	Y	Y	N	Stomach	40	China
Chinese CTR	ChiCTR2000029049	S, B	Y	Y	N	Colon, Rectum	50	China
Chinese CTR	ChiCTR2000032765	S, B	Y	Y	N	Bladder	1500	China
Chinese CTR	ChiCTR2000035441	S, B	Y	Y	N	Breast	30	China
Chinese CTR	ChiCTR2000036347	Ext B (Metastasis)	Y	Y	N	Prostate	100	China
Chinese CTR	ChiCTR2000037214	B	Y	Y	N	Pancreas	200	China
Chinese CTR	ChiCTR2000037237	Ext B (Metastasis)	Y	Y	N	Breast	90	China
Chinese CTR	ChiCTR-ONC-17011405	S, Ext B (Metastasis)	Y	Y	N	Breast	100	China
Chinese CTR	ChiCTR-OOC-17012057	B	Y	Y	N	Pancreas	50	China
Indian CTR	CTRI/2017/05/008512	B	Y	Y	N	Pancreas	20	India
German CTR	DRKS00021088	B	Y	Y	N	Pancreas	118	Germany
US NIH	NCT03283527	B	Y	Y	N	Esophagus	100	Netherlands
US NIH	NCT03307538	S, B	Y	Y	N	Bile duct	6	Netherlands
US NIH	NCT03429816	B	Y	Y	N	Stomach, Esophagus	40	Germany
US NIH	NCT03453307	S, B	Y	Y	N	Lung	100	China
US NIH	NCT03500068	Ext B (Metastasis)	Y	Y	N	Pancreas	30	Netherlands
US NIH	NCT03544047	S, B	Y	Y	N	Breast	50	China
US NIH	NCT03577808	S, B	Y	Y	N	Rectum	80	China
US NIH	NCT03764553	S, B	Y	Y	N	Esophagus	310	Netherlands
US NIH	NCT03890614	B (Bone Marrow)	Y	Y	N	Myeloid cells	40	USA
US NIH	NCT03896958	S, B	Y	Y	N	Various (solid)	1000	USA
US NIH	NCT03925233	S, B	Y	Y	N	Breast	300	China
US NIH	NCT03979170	S, B	Y	Y	N	Lung	50	Switzerland
US NIH	NCT04072445	S, B	Y	Y	N	Bile duct	28	USA
US NIH	NCT04261192	S, B	Y	Y	N	Head-neck	98	France
US NIH	NCT04278326	S, B	Y	Y	N	Vagina, Cervix, Penis, Oropharynx	50	France
US NIH	NCT04279509	B	Y	Y	N	Various (solid)	35	Singapore
US NIH	NCT04450706	S, Ext B (Metastasis)	Y	Y	N	Breast	15	USA
US NIH	NCT04469556	B	Y	Y	N	Pancreas	150	Canada
US NIH	NCT04526587	S, B	Y	Y	N	Breast	300	USA
US NIH	NCT04555473	B	Y	Y	N	Ovary	48	Italy
US NIH	NCT04561453	S, B	Y	Y	N	Biliary tract	20	USA
US NIH	NCT04587128	S, B	Y	Y	N	Colon, Rectum	110	USA
Netherlands CTR	NTR7521	S, B	Y	Y	N	Colon	150	Netherlands
Chinese CTR	ChiCTR1800017855	S, B	Y	Y	Y	Bones	200	China
Chinese CTR	ChiCTR1900024322	S, B	Y	Y	Y	Nasopharynx	160	China
US NIH	NCT03544255	S, B	Y	Y	Y	Pancreas	50	China
US NIH	NCT03655015	S	Y	Y	Y	Lung	50	USA

CTR, Clinical Trial registry; NIH, National Institute of Health; S, Surgery; B, Biopsy; Ext, Extended; Y, Yes; N, No.

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
