# Peer review of "Patient-Derived Tumor Organoids for Drug Repositioning in Cancer Care: A Promising Approach in the Era of Tailored Treatment"

_cancers, 2020, doi:10.3390/cancers12123636_

Round 1

Reviewer 1 Report

The Authors have made a careful review about the status of art on a new form of chemotherapy.

The paper is well written and it is easy to understand.
Patient derived tumor organoid presents several advantages which the Authors have well described in the paper, including te use of almost autolougous material in an innovative form of personalized cancer therapy.

Despite more than 60 clinical studies are ongoing, we still do not know the real potentials of this therapeutic approach.

Being the title of the collection "NEW TREATMENTS FOR COLORECTAL CANCER" we should inevitably accept discussions of new forms of therapy, which potentials need stiil to be  demonstrated.

The paper is an excellent review for the researchers involved in this field and for readers who do not about this innovative form on oncotherapy.

The Authors have described in details the potential advantages in research and clinical practice of a relatively new technique consisting in reproducing in vitro a 3Dimensional structure of a tumor.

These 3 Dimensional in vitro structures may be obtained from the patient culturing healthy cells genetically modified  or cells of the  tumor itself .

STRUCTURE OF THE PAPER

The Authors should re organize the paper in a more readable form, which often is obscured by their  evident enthusiasm about the technique,

The introduction should explain in detail the technique distinguishing between PDTO and PTD from genetically modified cells explaining pro and contras. In this context the introduction should be shortened avoiding comparison with standard in vitro cultures.

The Authors should avoid the continuous overemphasizing the advantages of the technique without specific  results which demonstrate that the potentials of the 3D in vitro culture had positive effects

The history of the technique described from line 139 to 170 should be summarized in a Table. The reader can be annoyed by the historical details.

Similarly the first applications of the technique may be more easily readable in a TABLE, eliminating the majority of the phrases from line 170 to 322, leaving just the most important concept and achievements.

CONCEPTUAL PROBLEMS TO RESOLVE

Inevitably the 3D in vitro culture offers many potentials, provided that we are not going to create an altered vision of the problem, which is probable and inevitable due to the time related changes in the development of a tumor and the many  factors involved including inhibitors, promoters, changing in chemical and physical condition of the environment as the tumor progresses.

These points should be addressed in two separate chapters:

1)Potential Advantages: this will allow to eliminate many of the confusing concepts expressed in the introduction. A summary table will be useful.

2)Potential Disadvantages: including altered environment, absence of many known and unknown factors facilitating tumor cell growth.

3)Limitations:  Each tumor is a selection of many abnormal clones of cells as we can see in the initial forms; as the cancer progresses  few selected clones proliferate representing the basis of the  advanced cancer. Thus, this evolution may be misunderstood or altered assuming that the tumor growth in vitro is similar to the tumor growth in vivo.

4)Apoptosis in vitro and in vivo: One of the major aspect of proliferating tumor is the continuous apoptosis of cells located in the central region of the tumor; this apoptosis may determine inflammation  and acidification which may stimulate cell growth through the production of IL1 IL2 TNF alfa and so on. Inevitably in vitro apoptosis may be more evident for the un physiological conditions around  the organoid .

To define possible difference in vitro and in vivo in cell apoptosis may be a major aspect to define.

5) Neo vessels formation Another significant aspect is related to neo vessel growth  and to the factors favoring this process, which  are difficult to quantify in vivo and inevitably in vitro.

All these aspects should be analyzed.

ADDITIONAL POTENTIALS AND QUESTIONS

-I accept the many potentials of the technique and the many possibility for improvement. One of the major problems I see is that genetic manipulation of health cells is inevitably complex, because we do not know the exact genes involve in many of the non genetically determined cancers. Which factors stimulate abnormal gene expression? What is the role of immunosenescence?

-in vitro testing of chemotherapy is important and may help to orientate the therapeutic approach . But how to assess the real effect of chemotherapy in vitro? The effect of chemotherapy may be confused with  the abnormal cell death related to malnutrition  in this complex in vitro organoid.

CONCLUSIONS.

I think appropriate to have a review chapter on this technique. I suggest the Authors to be a little more

objective explaining pro and contras of the technique. Inevitably in some situations the technique offers advantages and this should be specified , in another situation the technique may be misleading.

Overall the paper should be re-organized to make clear the above mentioned aspects of the Structure of the paper. 

The Author have done an excellent job. However a review is read by  “no expert” in the field and the “no expert” deserve clear messages, well exposed and easily understandable.

Author Response

We kindly acknowledge Reviewer 1 for the careful revision and positive comments which overall helped us to improve the quality of this manuscript. All the amendments and changes are highlighted in RED FONT within the updated version of the manuscript.

To answer IN DETAILS to Reviewer’s comments:

Q1 in STRUCTURE OF THE PAPER

A new Table 1 was generated and it contains all the detailed and more systematically organized information regarding the preclinical studies involving organoids described in: Sections 2, 3 and 5 (see Line 346).

Q2 in STRUCTURE OF THE PAPER and Q1 in CONCEPTUAL PROBLEMS TO RESOLVE

To better clarify the general advantages and disadvantages of the 3D-organoid technology in both basic and applied research, a new Figure 1 was prepared and thoroughly described within Lines 91-105.

As the Reviewer pointed out, such cellular models are grown in vitro with a very complex medium enriched in growth factors and nutrients, and moreover such cellular structures are 3-dimentional. 3D-organoids are a reductionist model to study tumor genesis and development in a dish. But this technique bears also limitations, including the fact that the cells are grown in a dish, far from the whole organism. Moreover, the 3D structure, if overgrown might determine an uneven diffusion of nutrients and this might bring to cell death, especially the inner core cells. All these important limitations were clearly reported in this newly added paragraph (lines 91-105) and new Figure 1.

Importantly, the advantages, the limitations and the potentials of PDTOs, the type of organoids representing the main focus of this review, are additionally discussed in Section 3, Lines 191-217 (that part was already present, but, thanks to the Reviewer’s suggestion, it was ameliorated to gain more clarity).

Q1 and Q2 ADDITIONAL POTENTIALS AND QUESTIONS

These general questions are very intriguing and for sure highlight the fact that 3D-organoids, like any other in vitro model, are a reductionist model.

This “reduction” can be seen as a double-edged sword. It is an advantage, because with the reduced numbers of variables we can control few conditions in the experiment, thus generating answers to our experimental questions more systematically. It is a disadvantage, because we might bear in mind that in vitro culturing could generate artefacts, such as the eventuality of unwanted cellular death (or senescence; see Lines 96-100).

That’s why, as reported in Lines 201-217 (and highlighted in Figure 3) the technology advancements (es., the genetic editing; the co-culturing with other cells - such as immune cells, stromal cells; the xenotransplantation in mice; the bioengineering) and trying to overcome the disadvantages, to add more “physiological/multiorgan” complexity to the 3D-organoids.

Despite that limitations, the very high amount of currently registered clinical studies involving the use of PDTOs in oncology, evidenced that 3D-organoids do not substitute clinical studies, but, despite the intrinsic limitations of an in vitro model, they can help the oncologist and the researcher, to: (1) screen compounds, (2) predict therapy outcomes, (3) tailor the therapy. For all these reasons, they are becoming a more and more useful complement in clinics.

Reviewer 2 Report

General comments:

Overall, the authors provide an interesting and comprehensive review in which current advances in organoid technology applied to oncology are discussed, describing a multitude of recent studies and approaches to this new available tool that is revolutionizing cancer research. They also include a very complete section on the history of organoids in translational oncology.

Minor comments:

I would recommend the authors to add some extra references:

- In Introduction, when comparing organoids with 2D cultures or animal models: Su-Jin Lee PMID: 33093266 and Lindsay J. Marshall PMID: 33092060.

- Line 99: When normal tissue-derived organoids are mentioned, apart from being used as a model of multistep transformation or initiation process (I suggest to include Jarno Drost PMID: 25924068 and Mami Matano PMID: 25706875), It would be interesting to mention more applications, such as toxicity studies (T. Shinozawa PMID: 33039464) and comparative studies between normal and tumor organoids and between different intestinal regions from the same patient (Fernández-Barral  PMID: 31306552 and Costales-Carrera PMID 32824266).

- Line 103: I suggest to add Jarle Bruun PMID: 32299813.

- Line 123: In relation to the isolation and culture of human organoids for the first time, Peter Jung PMID: 21892181.

- Line 201: Mikhail Nikolaev PMID: 32939089.

Typo errors:

- Line 47: “In Patient Derived Tumor Organoids represent…”, remove “In”.

- Line 145: “human heathy” should be “human healthy”

Author Response

We kindly acknowledge Reviewer 2 for the very nice general comments to our manuscript.

Regarding the minor comments, we thank the Reviewer for the helpful suggestions. We added ALL the 10 suggested references and integrated them with further description within the text (exactly where suggested). Now the manuscript gained more strength and completeness.

Regarding the typos, we amended the suggested ones and we carefully checked the whole manuscript for grammar mistakes. Corrections were made where needed.

All the changes are highlighted in RED FONT.

Reviewer 3 Report

Brief Summary

In this review, Vivarelli and co-authors provide an overview of how patient-derived tumor organoids are a promising platform for drug repositioning in cancer care. The manuscript revises the establishment of organoid models, their use for personalized medicine and their application in drug screening and drug development, including a perspective of extending the application of the tumor 3D models for the repurposing on cancer drugs for extending the treatment offer to cancer patients. 

Broad comments

The review addresses all the different patient-derived tumor organoids generated in the las 10 years, for what models there is already clinical data that supports the predictive value of the PDTOs and how they can be used for drug development and drug repurposing. Therefore, all the components for providing a proper overview of the subject are present in the manuscript. However, in some cases the argumentation is not connected and properly developed. In section 5, for example, a section focus in the use of organoids for drug repurposing, the authors mention in different paragraphs publications that described tumor heterogeneity (reference 16), Drug repurposing for COVID drugs (reference 91), screening activities in 384-well format (reference 21), and an international initiative to generate 100 PDTOs (reference 120) without a proper development and connection between the different publications.

Furthermore, in section 2, The importance of organoids in oncology, different organoid models are listed and whether PDTOs could be biobanked, genetic characteristics of the original tumor could be confirmed in the PDTO or what drugs were tested on these models is not consistently and clearly described for all the listed models. The generation of a Table summarizing these (and maybe other) relevant aspects of the PDTO’s applications would help to provide an overview.

The way different sections have been defined makes perfect sense, but as mentioned before, in each of the sections, some paragraphs are just list of descriptions without proper argumentation why are they mentioned. For example, in the section 5. PDTOs and drug repositioning: a new hope in cancer care, authors define in Figure 3 in what phases of drug repositioning PDTOs might play a relevant role and they define it in 2 out of the 4 phases described, without any argumentation or justification. I find that the selection of the phases where PDOs might play a relevant role in drug re-purposing need to be further developed in the text.

In general, authors have done a good job of synthesizing the literature, but I have two remarks. When referring, in section 2, all organoid models developed since 2009, I find relevant the citation of the animal organoid model if it is the foundation of the development for the corresponding human model. For example, this was indicated for mouse and human intestinal models developed in 2009 and 2011, respectively. However other animal-derived organoid models are refereed without including or describing the human counterpart. Since this review is focused on drug repurposing, only human-derived organoids, are relevant for this application. Therefore, I don’t see that much value referring only animal-derived organoids.

When describing key aspect for the development of organoid models (paragraph 5 of Introduction, lines 64-67), authors are referring already existing reviews (for example, references 9 and 10) rather than the original publications. The audience of the review will benefit by including as references original publications of the mentioned discoveries. Same for describing relevant drugs being repurposed, in section 5 Paragraph starting in line 349. Only a reference from a revie about repurposing drugs is indicated for 9 different drugs that re currently being repurposed.

This article would benefit from a close editing. Despite all relevant information is there, sometimes I found it difficult to follow the author’s argument due to the writing style used.

Regarding Figures and Table

In Figure 1, the authors are representing the two types of organoids that can be generated from either healthy or cancer tissue, however the two organoid models represented at left and right of the human shape are the same. It would be more visual to show purple (tumor) cells in the organoid model from the right. The figure will also benefit if figure legend describing the meaning of the different colours used to label the organoid cells is included.

Table 1 would benefit by adding the header of the columns in each new page.

In Figure 3, could authors indicate why they have defined the relevance of PDO in Compound validation and Clinical trial phases and not during Compound identification? 

Specific comments

There are some typo errors identified through the text:

Line 47, Second sentence should star w/o preposition "In".

Line 79, adjective do not include "s", therefore should be called organoid technology. Same in line 160, line 303, authors wrote "Organoids cultures" instead of "organoid cultures". Line 166 author wrote "organoids science" instead of "organoid science".

Line 142, Line 153, Line 158, Line 251, Line 265, Line 277, Line 303, authors wrote "Always" instead of "As well,"

Line 143, authors wrote "in turn able", might they meant "and turn able"?

Line 145, authors wrote "heathy" instead of "healthy"

Line 310, authors wrote "proprieties" instead of "properties"

In section 5., in second paragraph, line 349, authors indicate that “The compassionate use of drugs implies repurposing a known molecule, used for a primary non-oncology application, in order to offer an alternative, more effective, therapeutic option to cancer patients”, but what about repurposing cancer drugs that have been approved for one type of cancer that could benefit other type of cancers by re-testing them in different type of PDTOs? This option is not explored by the authors.

Author Response

We kindly acknowledge Reviewer 3 for the constructive comments. All the suggestions made were integrated in the text (RED FONT) and the manuscript gained in strength and clarity.

In particular, as suggested: (1) We improved the connections between reported findings in Sections 2, 3 and 5. (2) We implemented/elaborated more carefully argumentations where poor or missing within all the text. (3) All the typos and grammar mistakes suggested in the “specific comments” (and more) were amended accordingly. (4) According to the Reviewer’s suggestions, Table 1 was newly added and old Figures 1 and Figure 3 (now renamed respectively Figure 2 and Figure 4) were ameliorated (see below for details).

In details:

Section 5 (PDTOs and drug repositioning: a new hope in cancer care)

  1. Lines 382-385: Although it is out of the scope of the review describing the complete history of all the compounds currently in study or approved for repurposing in cancer care, as suggested, we included additional references to relevant ongoing preclinical and clinical studies.
  2. Lines 386-402: As suggested we improved the clarity and the contents of this part of the section. We described in more detail the whole process of drug repositioning, defining the 4 phases illustrated in Figure 4 (old Figure 3). Moreover, (new) Figure 4 was modified according to the text and now has gained in clarity: (1) each phase has been numbered; (2) each phase has a complementary box indicating the types of techniques used. Importantly, PDTOs are used in Phases 2 and 3, both including in vitro experimentation.
  3. Lines 408-412: The repurposing of anti-cancer drugs off label for other types of cancers has been thoroughly discussed.
  4. Lines 425-434 and 435-474: the specific part dedicated to PDTOs use in drug repurposing has been ameliorated. (1) a tight descriptive link in support of the schematic in Figure 4 has been generated; (2) the REFs 131-132-99-100-33-133 regarding findings specifically correlated with the development of a “screening platform” have been more smoothly interconnected and new REFs 134-135-136 have been added.

Section 2 (The importance of organoids in oncology) and Section 3 (History and applications of PDTOs in translational oncology)

  1. The Section 2 represents a general overview about organoid technology history and its specific application to the oncology field. While the specific part dedicated to Patient-Derived Tumor Organoids and their translational potential in oncology (including biobanking) is widely described in the following Section 3.
  2. As suggested, in both Sections 2 and 3 whole information about biobanking, genetics and drugs used in the studies described was integrated in the new Table 1 (line 346).
  3. As suggested, parts and REFs relative to exclusively non-human organoids studies have been removed from Section 2.
  4. Figure 2 (old Figure 1), as suggested, was ameliorated. In particular, an explanatory figure legend with normal and cancer cell features was added. Moreover, in line with the figure legend, the organoids graphic representations were ameliorated accordingly.

Round 2

Reviewer 3 Report

The authors have addressed all the comments made in the revision, so I don't have further requests.